# Bystanders’ Views on the Use of Automated External Defibrillators for Out-of-Hospital Cardiac Arrest: Implications for Health Promotions

**DOI:** 10.3390/ijerph18031241

**Published:** 2021-01-30

**Authors:** Susan Ka Yee Chow

**Affiliations:** School of Nursing, Tung Wah College, 31 Wylie Road, Homantin, Kowloon, Hong Kong, China; susanchow@twc.edu.hk

**Keywords:** cardiac arrest, automated external defibrillator, attitude, social responsibility, predictors

## Abstract

Despite the widespread availability of automated external defibrillators, not everyone is enthusiastic about using them. The aim of this study was to examine the reasons for not using an automated external defibrillator (AED) and predictors of the reasons. The study had a cross-sectional design using an online survey. Data were collected in eighteen districts in Hong Kong to be representative of the city. The questionnaire consisted of questions on demographics, knowledge and attitude towards AED use, reasons for not using AED, and whether the kind of victim could affect the decision of the bystanders. There was a high significant correlation between knowledge and attitude, with *r* = 0.782 and *p* < 0.001. Of the respondents, 53.3% agreed that the kind of victim would affect their willingness to operate an AED. A binary logistic regression model revealed that a higher education (OR 6.242, 95% CI: 1.827–21.331), concern about the kind of victim involved (OR 2.822, 95% CI: 1.316–6.052), and a younger age were significant predictors of worrying about taking on responsibility in using AED. Other than knowledge, other barriers included a desire to avoid legal liability, and the kind of victim they encountered. Life experiences in adulthood could possibly affect the social responsibility and influence the behaviors of adults to operate AEDs.

## 1. Introduction

Out-of-hospital cardiac arrest (OHCA) has long been a major cause of mortality worldwide. Given its high incidence and low survival rate, early recognition and intervention are essential. Some individuals suffering from cardiac arrest have died before reaching the hospital. The survival rate of OHCA victims in Hong Kong is suboptimal. Only 2.3% of patients were alive 30 days after the event or survived to be discharged from the hospital [1].

According to the American Heart Association, early recognition, cardiopulmonary resuscitation (CPR), and rapid defibrillation are essential links in the chain of survival outside of the hospital [2]. The most basic form of automated external defibrillator (AED) can be operated anywhere by trained bystanders using simple audio and visual commands. The European Resuscitation Council (2015) highlighted the importance of interplay between the bystanders who are able to provide CPR and timely implementation of AED [3]. Evidence from previous research and a systematic review with a meta-analysis showed that the early initiation of AED by a bystander before the arrival of Emergency Medical Services was significantly more likely to result in the patient’s survival to be discharged from hospital and with subsequent favorable functional outcomes [4,5].

There have been previous studies identifying barriers to using an AED. It was found that less than half of pedestrians were willing to use publicly available AEDs when they walked by Central Station in the Netherlands [6]. Although a study suggested that the recommended interval for refresher courses on AED and CPR skills is 7–12 months after the holding of the first training session to reduce skill decay [7], a lack of public knowledge, inadequate confidence, and an inability to find a nearby device restricted bystanders from using the device on the street [8]. Some students were found to be unable to identify CPR and AEDs and could not recall the location of the device in a student center [9]. Despite the widespread availability of the device, it was noted that not everyone is enthusiastic about using it, partly because of doubts about their knowledge and incompetence. In a study involving inductive in-depth thematic analyses of interviews, that using the AED voice prompt with a proper apparatus, having the ability to take charge of the situation, and feeling a moral obligation to act were drivers of participation in resuscitation attempts [10].

The unguided placement of AEDs was considered the main factor inhibiting the willingness of Hong Kong people to use the device [1]. While for people in Singapore, Korea, and Japan, a lack of knowledge, fear of doing harm, a lack of confidence in applying the skills, the feeling that it was better to call 911, the burden of taking responsibility, confusion and panic, and the difficulty of finding an AED were barriers to participating in efforts to save a life [11,12,13,14]. Although efforts have been paid off by nurses and healthcare team and television campaigns, there has been a significant increase in the number of people who have acquired the skills to use an AED; their willingness to perform chest compressions in accidents has remained unchanged [15,16].

Due to cross-national differences and economic status, the results of the above-mentioned studies cannot automatically be applied to a community with different characteristics. Moreover, the overseas studies did not specifically identify the predictors of unwillingness to use an AED. In our study, we seek to address the need to improve the implementation of AED. Our research questions are: What are the knowledge and attitudes of Hong Kong adults towards AED use in cases of OHCA? What are the reasons for the failure to use AED and what are the predictors of those reasons?

## 2. Materials and Methods

### 2.1. Study Design, Sampling, Setting, and Data Collection

A cross-sectional design was used in this study. Data were collected from residents of 18 districts in Hong Kong [17] to be representative of the situation in the city.

The sample size was based on a previous Korean study by Jung, Oh and Jeong [18], revealing the correlation coefficient between CPR attitude and knowledge, where *r* = 0.269, *p* < 0.001. By referring *r* to the formula developed by Hulley [19], an estimated sample size of no less than 123 subjects was required. With an estimated 20% drop-out rate, the final sample size needed to be approximately 147–150. Please refer to the formula below.

The standard normal deviate for
α = Za = 1.960

The standard normal deviate for
β = Zβ = 0.842
C = 0.5 *In[(1 + r)/(1 − r)] = 0.255
Total sample size = N = [(Zα + Zβ)/C]2 + 3 = 123

Quota sampling was adopted for data collection, with a proportional number of participants recruited from each district. The procedure for quota sampling is like convenience sampling, where the people in any subgroup are a convenience sample from that stratum of the population [20]. A participant was considered eligible if he or she was aged 18 to 64 years, a permanent resident who had lived in Hong Kong for at least 3 years or more, and was able to read Chinese. The exclusion criteria were an inability to read Chinese and having lived in Hong Kong for less than three years.

The questionnaires were delivered using online mode to the participants living in eighteen districts in Hong Kong. These eighteen districts incorporate the entire population area in Hong Kong. The population density per district varies from 825 (in remote areas) to 56,779 (densely populated area) per km^2^ [17]. The questionnaires were distributed through the community and social networks of the researcher and the student team.

### 2.2. Instruments

Our questionnaire consisted of questions on demographic characteristics and on knowledge and attitude towards AED. For AED knowledge, a set of validated questions from a previous local study [20] was adopted. The 10-item questionnaire was originally written in English [21] and had undergone backward and forward translation by Fan et al. [22]. Sample questions include: ‘Do you know the location of an AED nearest to your home or workplace?’, ‘Which one is the correct position for the placement of the AED pads? (With four diagrams shown to the respondents)’. With regard to the locations of AEDs, the score for this item ranged from 0–5, depending on how many locations the participants were able to identify. The total score of the scale ranges from 0–14, with a higher score indicating better knowledge.

For attitude towards AED, a scale in Chinese consisting of a set of 10 validated questions about CPR knowledge and attitudes among high school students was adopted [23]. The items are scored on a five-point Likert scale, from strongly agree to strongly disagree. A higher score indicates a more positive attitude towards the use of AED. Sample questions include: ‘The operation of AED should be the responsibility of healthcare professionals but not the laymen, ‘Learning AED is a complex task and is not suitable for ordinally citizens’, ‘Receiving AED is not necessary’. A few items are reversely phrased and required to score in reverse. The content validity index of the questionnaire was 0.954. A few phrases were changed from CPR to AED to make these items relevant to our study objectives. A pilot test was conducted prior to actual data collection. Due to modifications of the questions, there were 15 participants invited to review the questionnaire to determine whether the items are comprehensible to laymen and the duration to complete the entire set of questionnaires. To ensure the scale is having good stability, the reliability for test-retest was conducted by intraclass correlation (ICC), with ICC value equal to 0.997, which indicated excellent reliability [20].

### 2.3. Statistical Analysis

A statistical analysis was performed using IBM SPSS Statistics for Windows, version 26.0 (IBM Corp, Armonk, NY, USA). A descriptive analysis (frequency and percent) was used to describe the demographic data. The data were checked for normality to determine whether to use a parametric or non-parametric test, such as a Pearson or Spearmen correlation test, to determine the associations between the variables. Multiple linear regression and logistic regression were used to identify the predictors of attitude and worrying about taking responsibility to operate an AED on the scene, respectively. A *p*-value of <0.05 was considered statistically significant, based on the results of a two-tailed test.

### 2.4. Ethical Considerations

The study was approved by the Research Ethics Committee of the participating institute (NUR/SRC/20200106/020). Implied consent was used in the study. A detailed information sheet including the objectives of the study and the contacts of the principal investigator were provided to the participants. The participants were free to refuse to take part in the study and to withdraw from participation.

## 3. Results

A total of 175 questionnaires were distributed through social networks, and 158 questionnaires were returned. Eight questionnaires were discarded due to a large number of non-response items. The number of questionnaires that remained for analysis was 150.

### 3.1. Sample Characteristics

The participants ranged in age from 18–64 years old, with the largest number falling into the 18–24 age group. There were slightly more males (51.3%) than females (48.7%). The occupations of the participants varied, with 12.7% of them serving as healthcare workers; others were clerks, students, salespersons, and involved in other occupations. Most had completed a bachelor’s degree (38%). The majority of participants (70%) did not have a family history of heart disease. Forty-four per cent of the participants claimed that they had received CPR training, with only 30.7% reporting that they had received AED training. The respondents came from eighteen districts in Hong Kong. The highest percentage were living in the Central and Western district and Wong Tai Sin. Please refer to Table 1 for details.

### 3.2. Knowledge on AED

With regard to knowledge of AED, the scores for the 10 questions ranged from 0 to 14. The mean was 7.22 (SD 4.22). A certain proportion of participants (14.7%) had a very low score of 2. The scores of 66 (44%) participants were higher than the mean score of 7.22. The overall scores were not normally distributed.

Regarding the compression rate when performing CPR, only 41.3% of the respondents were able to give a correct answer of pressing down at a steady rate of 100–120 compressions a minute. Overall, 43.3% of the participants responded correctly that with a delay to apply the AED in one minute, the survival rate would decrease to 7–10 percent. While 56.0% of the respondents have knowledge about someone without a heartbeat for longer than 5 min, the brain could suffer permanent brain damage. On the proper position to place the electrodes, 46.0% of the participants gave the correct answer. Less than half of the respondents, 42.7%, were aware of the location of AEDs nearby their homes and workplaces.

### 3.3. Attitude towards Using an AED

Concerning attitude towards using an AED, the scores ranged from 16 to 50 for 10 questions. The mean score was 33.47/50 (SD 7.53). More than half of the participants (53.3%) had a score of below the mean. The scores for attitude were normally distributed.

Among the respondents, 52.7% responded that learning AED is a complex task and is not suitable for ordinary citizens. Similarly, the same percentage of participants, 52.7%, agreed, and had no particular concern that operating the AED should be the responsibility of healthcare professionals. If the victim is the family member, 79.3% agreed to use an AED for the victim. Among all the participants, 78% agreed to further promote AED education and training in the community.

### 3.4. Correlation between Knowledge and Attitude towards Using an AED

A Shapiro–Wilk test was used to examine the normality of the measurement scales, including the knowledge and attitude tests. Data are considered normally distributed when the significant value is greater than 0.05 [24]. As the knowledge score was not normally distributed, a Spearman rho correlation was used to determine the correlation between knowledge and attitude. The r was 0.782 and *p* < 0.001. The results indicated a high correlation between the two variables.

### 3.5. Reasons for Not Operating an AED on the Scene

On the reasons for not operating an AED on the scene, the respondents could choose more than one answer. One hundred and five respondents chose ‘Worry about taking on responsibility’, 80 chose ‘Incompetent and without AED training’, while 73 selected ‘Worry about suffering from infectious diseases’. Please refer to Figure 1 for details.

On whether the kind of victim would affect the bystander’s tendency or decision to operate an AED, more than half of the respondents, 53.30%, responded ‘Yes’. Among those 53.30%, the respondents would be most likely to offer help to a family member in the case of a sudden cardiac arrest, and less likely to offer assistance to a complete stranger, such as any pedestrians. Please refer to Figure 2 for details.

### 3.6. The Predictors for Worrying about Taking Responsibility to Operate an AED on the Scene

Based on the previous reasons for not operating AED on the scene, a binary logistic regression analysis was performed to identify the predictors of ‘Worry about taking on responsibility’. For the dependent variable, an answer of ‘Worry’ equals to 1 and 0 indicates not worry. The independent variables were gender, level of education, age group, whether the kind of victim involved could affect the decision of the bystanders, and knowledge and attitude scores. A selection of these dependent variables was based upon previous studies [10,11,12], a systematic review [25], Shams al’s study [26] on how demographics are affecting people performing CPR, as well as the high correlation between knowledge and attitude scores in our results. The results of the binary logistic regression indicated that there were significant associations between education, age, and the kind of victim involved. Table 2 showed that people with a higher level of education, bachelor’s degree or above comparing with secondary school education (OR 6.242 (1.827–21.331), and those who had a concern for the kind of victim involved (OR 2.822, 95% CI:1.316–6.052), were more likely to worry about taking on the responsibility of using an AED, while those of older age compared with the youngest group (OR 0.095, 95%CI: 0.017–0.520; OR 0.149, 95% CI:0.029–0.769) were less likely to worry. Please refer to Table 2 for a model of the analysis.

### 3.7. The Predictors of Attitude in Using AED

Based on the results of the correlation analysis between knowledge and attitude, a multiple regression analysis was performed to examine the predictors for the attitude score. The total score on attitudes was used as the dependent variable, while the independent variables were: the total score on knowledge, demographics, and whether the kind of victim involved affected the use of an AED. The knowledge score and the kind of victim involved were found to be significant predictors of attitude, with adjusted *R^2^* = 0.637. Having more knowledge (β = 0.699, 95% CI = 1.051–1.430, *p* < 0.001) was positively corelated with a good attitude. On the other hand, while the kind of victim involved was a significant predictor, those respondents who answered ‘Yes’ to the question had a less positive attitude (β = −0.2, 95% CI = −4.614–1.403, *p* < 0.001). A model of the regression analysis is shown in Table 3.

## 4. Discussion

It has been well documented that knowledge and attitude play an integral role in determining a person’s readiness to participate in different aspects of health promotion activities, including implementing CPR and operating an AED.

The aim of this study was to examine the respondents’ knowledge, attitude, and the factors involved in not operating an AED on the scene. It has been well documented that knowledge and attitude play an integral role in determining a person’s readiness to participate in different aspects of health promotion activities, including implementing CPR and operating an AED. Although similar studies have been conducted in Hong Kong, they lacked representativeness as the participants were recruited only via convenience sampling in three locations around the shopping malls and subways [22]. The discussions were mainly descriptive without examining the correlations nor identifying the predictors of variables. The results showed that 65.8 % and 85.3% of the respondents had no training in first aid and the use of an AED, respectively. The public knowledge of AED was low as well.

Our study found a high positive correlation between knowledge and attitude. This finding was corroborated by studies of people from all walks of life. For example, a training program in Singapore was able to impart new information and skills on CPR and AED, resulting in an improved attitude on the part of school children [27], while a large majority of teachers and students in Japan with AED knowledge indicated that they would certainly be willing to operate an AED [28]. Furthermore, a significant correlation was found between knowledge and attitude towards AED and CPR among university students in Korea [14]. By contrast, some US university students were not comfortable with using AED without assistance [9]. To further examine the discrepancies, a recent systematic review by Smith [25] illustrated that while people were willing to obtain the relevant skill, a majority of those were not comfortable about using it, due to fears of legal liability and of causing harm to the patient. Most importantly, the available evidence was of inadequate quality, and the great heterogeneity in the mode of conducting surveys threatened the external validity of the findings.

Regarding the reasons for why bystanders would not apply an AED at the scene, the prominent reason in our study was the worry about taking on responsibility. A majority of our respondents would offer help to their family members, while only a minority would give assistance to complete strangers. The results of the logistic regression further confirmed that bystanders would be concerned about the kind of victim involved, and that those who were older were less likely to worry about taking on responsibility in using an AED. Finally, the linear regression showed that the kind of victim involved was a significant predictor, while those respondents who answered ‘Yes’ to that question had a less positive attitude towards using an AED.

The fear of harming an OHCA victim and the possible liabilities involved are the main reasons why bystanders would hesitate to perform AED. The consequence of this reluctance to help a victim who is a stranger is a poor OHCA survival rate [29,30]. This phenomenon is also seen in the performance of CPR, as CPR shares a similar characteristic to AED in saving OHCA victims. With respect to legal liability, according to provision 1714.21(d) of the California Civil Code, an uncompensated Samaritan, whether a person or an entity, that acquires an AED for emergency use is not liable for any civil damages resulting from any acts or omissions in the rendering of the emergency care by use of an AED. California also requires health clubs to have at least one AED [31]. In Hong Kong, many organizations have been organizing courses to educate the general public on AED or CPR. Nevertheless, the Hong Kong government to promote bystander intervention is limited. Up to the present, there is no legislation in Hong Kong providing an exemption of rescuers from legal liabilities that might be incurred in performing first aid [32]. A local study also highlighted the fact that it is common in Hong Kong for healthcare professionals to act as volunteer rescuers, but the legal risk related to resuscitation should not be underestimated. Hence, the government needs to consider putting legal protections in place for first-aiders in order to increase the willingness of the public to use AEDs [33]. Although many studies have suggested that a Good Samaritan’s Law could enhance the public’s willingness to use AED or CPR because of the exemption of any civil or criminal liability when any emergency medical service is provided to a patient, arguably the effectiveness of having such a law is not high. In Korea, even though a Good Samaritan’s Law has been in effect since 2008, only 40.2% of people have heard of such a legislation [34]. Clearly, this law is not well known. Therefore, the government should not simply enact new legislation to protect bystanders who try to help someone they believe to be in danger of harm, but should also make an effort to widely publicize the laws, to ensure that citizens are familiar with the established laws.

An interesting finding in our study was that, with an increase in age, people tended to be less worried about taking on responsibilities. A previous study revealed there are no stable developmental patterns in adult life with regard to social responsibility. Rather, the specific pattern depends upon a person’s sex [35]. Regarding the age effect, Schaie found that the measure of social responsibility appeared to have the highest correlation with years of education and intelligence during the period of middle age. Later in life, the sense of community responsibility seems to be moderate, with the result that other factors might become more important in determining attitudes towards a person’s social obligations [36]. Even if a person is educated, his/her life experiences, whether positive or negative, may explain why an experienced individual is happy or less happy to shoulder social responsibility. Therefore, further evidence based on social responsibility and age is needed to come to a robust conclusion about the association between age, level of education, and the willingness to operate AEDs in the community. Lastly, the promotion of society responsibility and volunteer experiences is worthy of attention in every society. To further promote health and embedding a board and positive health concept, general and health education is not simply to educate competence, but the goals are to support caring and responsible citizens in family, workplaces and community.

This study was able to address the previous gap in knowledge on the subject. A key strength of this study was the inclusion of a few questions investigating the reasons for not using an AED on the scene and whether the kind of victim could determine whether the bystanders chose to operate an AED on the scene.

### Limitations

There are several limitations in the study which need to be considered. First, because of the cross-sectional design of the study, causal relationships between the study variables could not be determined. Second, the data were collected in one city which lacks the generalizability to other population group. To get more detailed results, a comprehensive analysis including mixed methods, including interviews of the participants, are warranted in future studies.

## 5. Conclusions

In this study, a high correlation was observed between knowledge and attitude towards using AEDs. Other than knowledge, our study expands the evidence to include barriers to the implementation of AED by bystanders in cases of OHCA determined by demographics such as age and education, avoidance of legal liability, and the kinds of victims the bystanders encountered. A person’s life experiences in adulthood might have an impact on that individual’s sense of social responsibility with regard to operating AEDs. These new insights not only provide an additional research area, but inform the delivery of nursing and healthcare education to the public from a sociological perspective.

## Figures and Tables

**Figure 1 ijerph-18-01241-f001:**
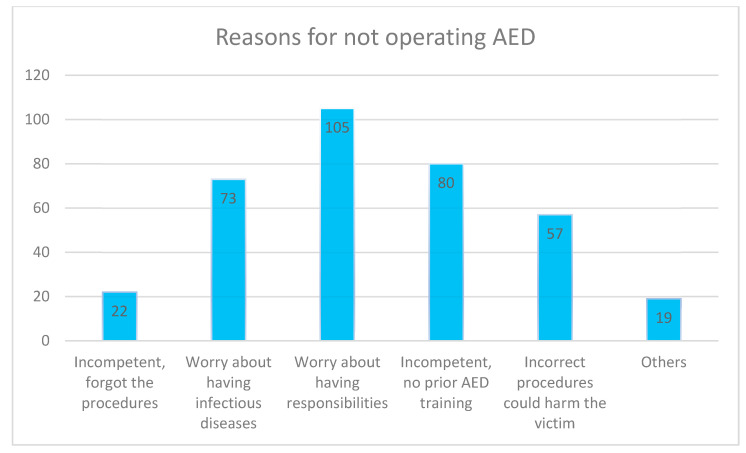
Reasons for not operating AED. The respondents could choose more than one answer to this question.

**Figure 2 ijerph-18-01241-f002:**
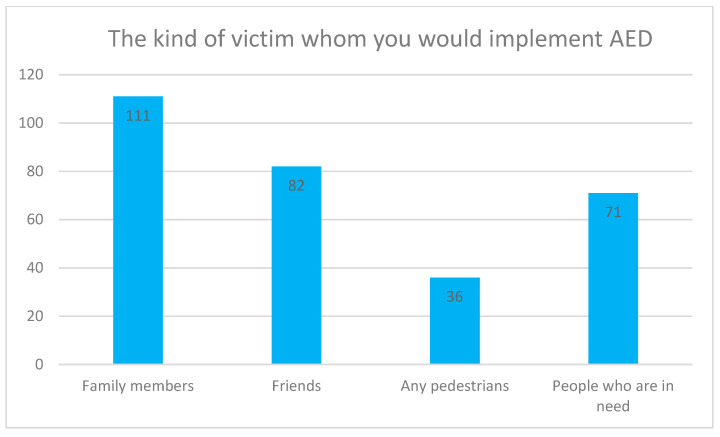
The kind of victim on whom you would implement AED. The respondents could choose more than one answer for this question.

**Table 1 ijerph-18-01241-t001:** Demographic characteristics of the participants (N = 150).

Characteristics	No. of Participants (%)
Gender	
Male	77 (51.3)
Female	73 (48.7)
Age (years)	
18–24	66 (44.0)
25–39	60 (40.0)
40–64	24 (16.0)
>65	0 (0.0)
Level of education	
Secondary school	38 (25.3)
Post-secondary school	21 (14.0)
Associate degree	34 (22.7)
Bachelor’s degree or higher	57 (38.0)
Occupation	
Healthcare professional	19 (12.7)
Teacher	13 (8.7)
Others	118 (78.7)
First Aid Certificate	
Yes	66 (44.0)
No	84 (56.0)
Previous AED training experience	
Yes	46 (30.7)
No	104 (69.3)
History of heart diseases in family	
Family members have heart disease	45 (30.0)
Family members do not have heart disease	105 (70.0)
District of residence	
Central and Western	13 (8.7)
Wan Chai	8 (5.3)
Eastern	9 (6.0)
Southern	9 (6.0)
Islands	3 (2.0)
Yau Tsim Mong	10 (6.7)
The Kowloon city	2 (1.3)
Wong Tai Sin	13 (8.7)
Sham Shui Po	6 (4.0)
Kwun Tong	8 (5.3)
Tsuen Wan	8 (5.3)
Kwai Tsing	6 (4.0)
Sai Kung	8 (5.3)
Sha Tin	11 (7.3)
Tai Po	7 (4.7)
North District	9 (6.0)
Tuen Mun	8 (5.3)
Yuen Long	12 (8.0)

**Table 2 ijerph-18-01241-t002:** Binary Logistic Regression Analysis on the predictors of ‘Worry about taking on responsibility’.

Variables	Odds Ratio (95% CI)	*p*-Value
Attitude score	0.933 (0.848–1.027)	0.158
Knowledge score	1.138 (0.963–1.344)	0.130
Gender		
Male	0.899 (0.393–2.056)	0.800
Female	1.00 ^Ref^	
Concern about the kind of victim		
Yes	2.960 (1.187–7.384)	0.020 *
No	1.00 ^Ref^	
Age Group		
18–24	1.00 ^Ref^	
25–39	0.095 (0.017–0.520)	0.007 **
40–65	0.149 (0.029–0.769)	0.023 *
Education		
Secondary school	1.00 ^Ref^	
Post-secondary school	5.779 (1.656–20.127)	0.006 **
Associate Degree	0.752 (0.205–2.754)	0.667
Bachelor’s Degree or above	6.242 (1.827–21.331)	0.003 **

* *p* < 0.05, ** *p* < 0.01. ^Ref^: this category is for reference with the variables below.

**Table 3 ijerph-18-01241-t003:** Linear Regression Analysis for predictors of positive attitude in using AED.

Variables	Regression Parameter (Beta)	*t*-Statistics	*p*-Value
Age Group	0.21	0.408	0.684
Gender	0.51	0.984	0.327
Education	0.84	1.487	0.139
Knowledge score	0.699	12.946	<0.001 *
Concern about the kind of victim	−0.200	−3.703	<0.001 *

* *p* < 0.001.

## Data Availability

The data presented in this study are available on reasonable request from the corresponding author.

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
