# Peer review of "Bystanders’ Views on the Use of Automated External Defibrillators for Out-of-Hospital Cardiac Arrest: Implications for Health Promotions"

_ijerph, 2021, doi:10.3390/ijerph18031241_

Round 1

Reviewer 1 Report

The Author improved the previous version of the manuscript. 

The topic is interesting, as methodology I think could be improved citing the numeber of inhabitants of the eight considered district.

Author Response

I guess there is a typo for the comment.

There are eighteen districts in Hong Kong but not eight considered district. I added the below explanations in line 91-93 to make it clear for the readers from overseas. "These eighteen districts incorporate the entire population area in Hong Kong. The population density per district varies from 825 (in remote areas) to 56,779 (densely populated area) per km2. " 

Reviewer 2 Report

It is a timely and interesting study.

I am not surprised about the results.

Resuscitation cannot be improvised by people who lack of knowledge, training and willingness to help.

A negative outcome is not infrequent even in a hospital environment and is related to many factors.

It is unrealistic to expect "the magic bullet" outside a hospital environment.

The article is fluent and well written although it requires further review of the English language.

Author Response

The entire paper has been reviewed thoroughly again for the English language. The revisions were lighted in red for easy references.  Thank you.

This manuscript is a resubmission of an earlier submission. The following is a list of the peer review reports and author responses from that submission.

Round 1

Reviewer 1 Report

I would have to say that this study lacks significant impact on this field. Novel findings or approach cannot be found out only investigating in the area of Hong Kong. Previous study which was more validated already demonstrated the barriers to applying life-saving skills (Ong ME, et al, Resuscitation, 2013, Shams A, et, Medicine, 2016…). The results would not be surprising. Please see the following specific comments.

  1. Please specify the novel findings or approach in this study which are totally different from previous studies.

  1. I am wondering why the author selected participants via a convenience sampling not random sampling. The level of bias would be extremely high that sample size calculation would be meaningless. I am pretty unsure about the target the author was focusing on, which were mixed with healthcare professional, teacher, and others.

  1. Every single question in terms of demographic characteristics, knowledge and attitude towards AED should be presented.

  1. The independent variables should be addressed in the subheading of statistics, and these should be included based upon the previous studies or potentially relevant factors.

Reviewer 2 Report

The suggested manuscript is a text describing a study and its results on the following topic :

Which factors are affecting the choice of bystanders to use or not to use an AED when in presence of a cardiac arrest.

The study has been performed based on questionnaires collected via social networks.

An introduction allows to understand the topic and its context, including possible studies overlapping with the presented work.

Description of the study design is performed and presents some explanations about sample size calculation and how statistics are performed. Both parametric and non-parametric correlation tests are disclosed to be used.

Paragraph results presents the compiled data extracted from all selected participants.

A paragraph is dedicated to discussion trying to determine the factors that are linked to attitude of possible bystanders to cardiac arrest.

Conclusion is placing the summarizing the results of the study.

The updates that should be provided are listed below:

In general, it is necessary to use the same text when the author refers to the same concept. For instance, it is mentioned “the kind of OHCA victim” (line 204) and figure 2 mentions “on whom would you implement AED”. Using the same wording would lead to easier reading. This is especially true because on line 221 it is mentioned “those who had concern for the kind of victim involved”, which is another description of the same population.

  • “Materials and Methods” / line 101:

It is mentioned a “test-retest reliability analysis”, with some results further (line 104) indicating a correlation of 0.994. It might be clearer to deliver these information in the “results” paragraph. Furthermore, it is not clear if the test has been applied to all set of question or only to some updated question (change from CPR to AED).

  • “Results” / line 123-125:

It is said that questionnaires were distributed through social networks.

It is not disclosed how the focus was done on the target population from these network. Since there is a possibility that the respondents would be people which are friends (friends as social network definition) from the authors, it should be disclosed. Indeed, it might be surprising to have only 175 answers on a large city like Hong-Kong. Are the respondents selected, based on their experience of a cardiac arrest?

Line 196 as title is duplicated with line 212.

In general this paragraph is unclear, mixing topics “reasons for not using an AED” with “on whom would you implement an AED”.

Line 205 : 53.3% of respondents chose “Yes”. Does that mean that on figure 2, respondents choosing “Anybody” did answer “No” to the question ? In that case 53.3% should fit with the following computation : (150-71) / 150, and it does not. Perhaps, definition of "Anybody" might be problematic : is it meant "nobody" ?

Line 214-215 “Worry about taking responsibility”, for the choice “0 otherwise”, the way it is disclosed means that this population is composed of recipients answers “Not worried about responsibilities” and “No answer to the question by the recipients”

Line 233-235 : Should be more accurately explained : “Kind of Victim” : “Yes” / “No”

Table 2: Line 238 : There is probably a typo error (written 29 instead of 39).

Table 3: Line 258 is mentioned “Age” whereas it seems to refer to “age group”.

Everywhere in the text: Please correct OCHA instead of OHCA.

Reviewer 3 Report

The topic is interesting, but it needs to be improved.

All the manuscript

1) correct the abbreviation for Out-of-hospital cardiac arrest the correct one is OHCA.

Introduction

2) cite in the introduction the guidelines of  The European Resuscitation Council (ERC) about the treatment of SCA.

Materials and Method

3) I think it is relevant to know if in HK there is a law about the positioning of AEDs in public/private areas. It is possible to know how many AEDs are in the district analyzed by the Author and also the population so we can have a ratio inhabitants per AED.

4) page 6 line 212 should be a new sub paragraph 3.5 ...

5) page 7 line 246 should be a new sub paragraph 3.6 ...

Discussion

6) page 7 line 268 - 271: the Author cite a previous study in HK should be done a comparison between those results and the present study.

7) page 8 from line 296: the author explain laws in California and in Korea but it is necessary to focus on HK and should explain if there is a similar law or not and eventually describe the content.